# Using multiple criteria for redesigning habitat corridor plans for Giant Pandas

Yixin Diao [ID][1,2], Yue Weng[1], Qianqian Zhao[1], Xiangxue Hu[2], Xiaofeng Zhang[3], Bojian Gu[1], Yihan Wang[1], Zhuojin Zhang[1], Fang Wang[1]*

1 Ministry of Education Key Laboratory for Biodiversity Science and Ecological Engineering, Coastal Ecosystems Research Station of the Yangtze River Estuary, Institute of Biodiversity Science, School of Life Sciences, Fudan University, Shanghai, China, 2 College of Forestry, Guizhou University, Guiyang, China, 3 Shaanxi Provincial Administration of Giant Panda National Park, Xian, China

* wfang@fudan.edu.cn

## Abstract

One key factor to long-term success in conservation planning is to allocate limited resources to the most critical locations and ensure the effectiveness of contemporary management plans under future conditions. Here, we proposed an innovative framework to quantitatively prioritize conservation actions for a vulnerable species, giant panda (*Ailuropoda melanoleuca*), and use it to identify Priority Habitat Corridors (PHCs) to address the threat of habitat fragmentation. We focused on the newly established Giant Panda National Park (GPNP), and combined field data with remotely-sensed landscape and anthropogenic metrics to evaluate the feasibility and effectiveness of potential habitat corridors. We first conducted habitat and corridor modeling and identified 72 candidate corridors. We then conducted gap analysis and applied multiple criteria to prioritize candidate habitat corridors. We identified 34,486 km$^2$ of suitable habitat for giant pandas inside the GPNP and six PHCs that merit the highest priority because they were located at optimal elevation range with adequate forest coverage, had the capacity to connect multiple habitat patches, and were relatively feasible based on their conservation and anthropogenic interference status. Using an integrated modeling approach with biophysical, biotic, and anthropogenic criteria, our work provides a generic methodology to prioritize conservation actions which support future management for giant pandas and other wildlife species.

## Introduction

Establishing protected areas, including nature reserves and national parks, is considered one of the most effective measures to conserve endangered species and their habitat [1,2]. Theoretically, protected areas are designated based on long-term goals to prohibit direct (e.g., poaching and deforestation) and indirect (e.g., noise and pollution) threats to biodiversity within their boundaries [3,4]. However, different protected areas vary greatly in their capacity to protect wildlife and their habitat [5],

**Data availability statement:** All relevant data are available from the Figshare repository (DOI: https://doi.org/10.6084/m9.figshare.28822028).

**Funding:** We thank the staff of the Shaanxi Forestry Department who helped us in collecting environment information. We are very grateful to Dr. McShea for his advice in conducting this research. The project was supported by the National Natural Science Foundation of China (31971537) and Qiankehe [ZK] (092). The funders had no role in study design, data collection and analysis, decision to publish, or preparation of the manuscript.

**Competing interests:** The authors have declared that no competing interests exist.

and many threatened species are still inadequately represented within the protected area system [6–8]. Currently nature reserves cover 10–15% of the terrestrial sphere [9,10], but a challenge to their further expansion is that many countries lack the necessary resources [11,12]. Half of world's protected areas have suffered significant deterioration and biodiversity loss during the past 30–40 years [13], among which even successful ones imperiled by eroding political support and a global trend of "downgrading, downsizing and degazettement" [14–16]. As conflicts between urgent conservation needs and insufficient resources remains, frameworks to better allocate limited resources to the most critical areas are thus essential to maximize successful conservation outcomes.

Giant panda (*Ailuropoda melanoleuca*) is a great example of a species facing severe habitat fragmentation that would benefit from a framework to identify Priority Habitat Corridors (PHC) and restore habitat connectivity [17]. To prevent the species' extinction, China established its first nature reserve for giant pandas in 1965 [18], and added 66 nature reserves during the subsequent fifty years, covering a total area of 33,562 km$^2$ (State Forestry Administration of China, 2017). While the rapid growth of nature reserves has slowed since 2015, severe habitat fragmentation continues to threaten the viability of isolated populations [19–22]. Establishing corridors to connect isolated populations has been advocated since 1980 [21] however, few of the proposed corridors were effective, and many marginal populations are still losing habitat as well as population numbers [23]. A major drawback hindering successful corridor establishment is the lack of systematic methodology to identify PHCs among the large number of candidate sites. Most previous studies have either focused on a specific corridor and ignored the overall landscape [24] or proposed too many candidate sites to be accomplished with limited resources [25,26]. An opportunity to develop such a framework is the newly announced Giant Panda National Park (GPNP), with its major goal to better coordinate conservation actions among existing nature reserves (State Council Information Office of China, 2017). The GPNP has an area of approximately 27,134 km$^2$, and spans three provinces of Sichuan, Gansu, and Shaanxi (State Council Information Office of China, 2017). A major challenge in managing such a complex conservation network is to restore habitat connectivity across administrative boundaries under the pressure of economic growth [27,28]. A priority framework that can quantitatively measure the importance, feasibility, and potential effectiveness of candidate corridors is thus vitally important to a successful conservation plan for the species.

Here we propose the first framework, which used integrated modeling approach with biophysical, biotic, and anthropogenic criteria to identify PHCs in and around GPNP. The framework would have direct implications to the conservation of giant panda and sympatric species, as well as to other conservation networks where habitat connectivity restoration is in critical need.

## Method

### Designing a priority framework

We first designed a framework based on previous studies (Fig 1; [29–36]), then we applied the framework to identify PHCs in GPNP following a three-step procedure:

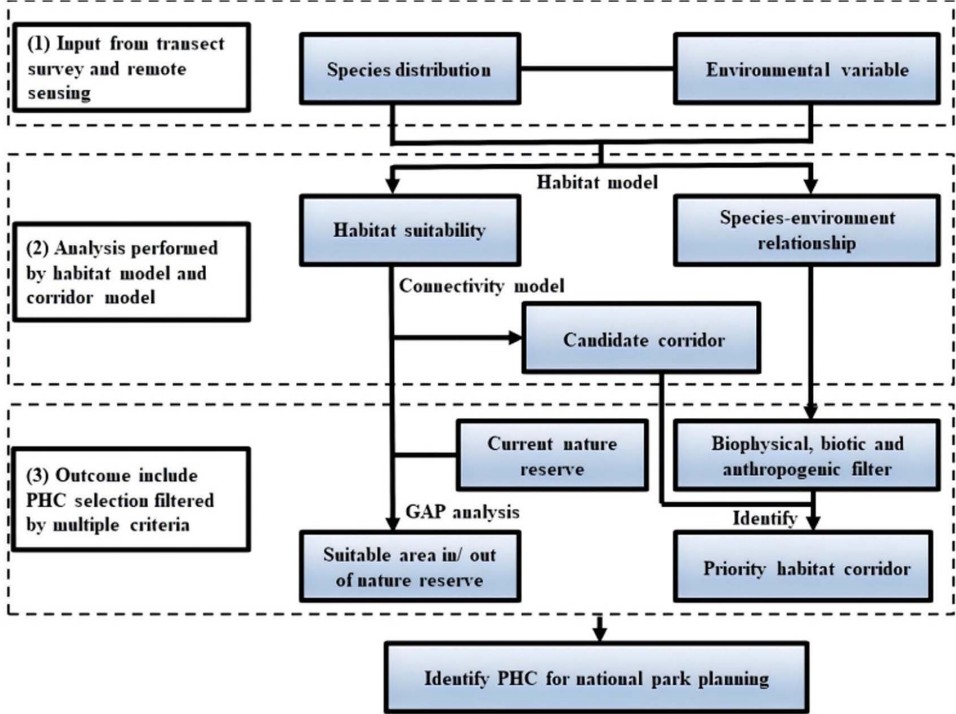

**Fig 1. Framework to identify Priority Habitat Corridors and support Giant Panda National Park conservation planning.**

(1) collect species, environmental, and socio-ecological data; (2) construct habitat and corridor models to identify candidate PHCs; and (3) use biophysical, biotic, and anthropogenic criteria as filters to select PHCs based on their importance, potential effectiveness and feasibility [37–41].

## Study area

We used the GPNP and its surrounding historical giant panda distribution and the administrative boundaries of Sichuan, Shaanxi, and Gansu Provinces as our study area (Fig 2). The study area encompasses six mountain ranges (Qinling, Minshan, Qionglai, Greater Xiangling, Lesser Xiangling, and Liangshan Mountains), with an elevation range from 232–5,787 m, covering all known giant panda ranges [20,42].

There are 67 giant panda nature reserves in this area. While most nature reserves are well-management with minimum direct anthropogenic disturbance (e.g., poaching, illegal timber harvest), outside the nature reserves are major agricultural areas, industrial development zones, and townships connected by roads and railways that may form barriers to movement [43].

## Data collection

We obtained giant panda distribution data collected during the 4th National Giant panda Census (2011–2013) (State Forestry Administration 2015). During this census, field staff recorded giant panda signs (i.e., feces, dens, foraging sites, and footprints) at approximately 8,000 locations along 1,076 transects (Sichuan Forestry Department 2015; State Forestry Administration 2015). Giant pandas were considered present where any evidence was found. To construct statistical models requiring both species' presence and absence values, we used reported giant panda home range size (approximately 5 km², see [44]) as threshold, and generated random pseudo-absent locations 2 km away from the presence points along the transects [45].

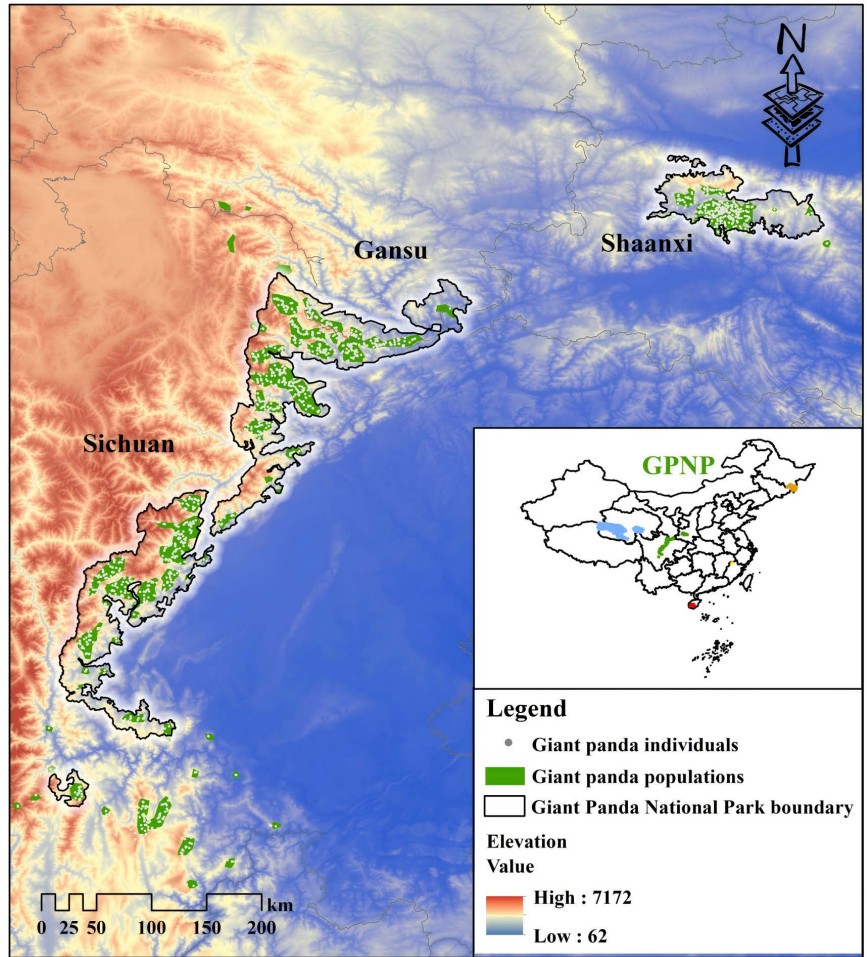

**Fig 2. Giant Panda National Park.** There are 1864 giant panda individuals inside GPNP or around it. Based on China's fourth National Giant Panda Survey, giant pandas are distributed 33 isolated populations in the wild, 18 of which will be included within the boundaries of GPNP. There are 5 national parks announced by Chinese government in the Conference of the Parties 15, Kunming, including: Giant Panda National Park (leaf green); Sanjiangyuan National Park (yogo blue); Hainan Tropical Rainforest National Park (mars red); Wuyi Mountain National Park (solar yellow); Northeast China Tiger and Leopard National Park (seville orange).

We reviewed previous literatures on giant panda-environment associations [23,46–48]and selected nine variables that were potentially important for their habitat selection (Table 1). We obtained nature reserve boundaries from the Sichuan, Shaanxi, and Gansu Forestry Department [42,49,50]. Geo-referenced data of residential areas and roads were obtained from the State Forestry Administration [49]. Distance to road, nature reserve, and residential areas was calculated in ArcGIS 10.2 using the *Distance* tool [23,48,51,52]. For topographical variables, we used 30-m resolution DEM (ASTER 2009) [53] to delineate the elevation, slope, terrain ruggedness and aspect of each raster grid (we brought all data layers to same scale, 300 m x 300 m, see [54]). We obtained vegetation composition from GlobCover 2009 data [55,56], and aggregated vegetation composition into four types based on giant panda habitat preference (coniferous, mixed, broadleaf, or non-forest) [23]. We used Variance Inflation Factor (VIF) as a descending dimension algorithm to examine multicollinearity among variables [57]. We found no evidence of significant multicollinearity as all variables had VIF < 5 [58,59], and all variables were retained in the following analysis (Table 1).

**Table 1. Variables used for giant panda model construction of each variable.**

| Name | Description | Source | VIF * |
|------|-------------|--------|-------|
| Elevation | Numeric (km) | ASTER 2009 | 1.51 |
| Terrain ruggedness | Numeric | ASTER 2009 | 1.49 |
| Slope | Numeric (°) | ASTER 2009 | 2.07 |
| Aspect | Categorical (north, east/ west, south) | ASTER 2009 | 1.00 |
| Forest composition | Categorical (nonforest, broadleaf, mixed, conifer forest) | Bicheron et al. 2008 | 1.34 |
| Nature reserve | Categorical (no, yes) | Forestry Departments | 1.10 |
| Distance to nature reserve | Numeric (m) | State Forestry Administration 2021 | 1.08 |
| Distance to residential area | Numeric (m) | State Forestry Administration 2021 | 1.69 |
| Distance to road | Numeric (m) | State Forestry Administration 2021 | 1.57 |

*VIF: Variance Inflation Factor for collinearity test

## Modeling giant panda habitat

We randomly selected 70% giant panda locations for model training and used the remaining 30% to assess the discriminative ability of different model algorithms [60]. Six commonly used algorithms (Classification Tree Analysis; Flexible Discriminant Analysis; Generalized Boosting Model; Generalized Linear Models; Multivariate Adaptive Regression Splines; and Random Forest) were constructed and later ensembled based on cross-validation results for more robust results [61]. The procedure was repeated 10 times to eliminates any models with average AUC (Area Under the Receiver Operating Characteristic Curve) or TSS (Total Sum of Squares) value lower than 0.85 [60].

To transform the continuous habitat suitability predictions to binary habitat/ non-habitat interpretations, a sensitivity and specificity-combined approach was used to calculate the threshold, which can maximize the sum of model sensitivity and specificity [62,63]. Our habitat models were constructed in R using 'biomod 2' package [64].

We overlapped our suitable habitat prediction with current nature reserve boundaries, and conducted GAP analysis to evaluate the effectiveness of current conservation network on covering suitable habitat for giant pandas [65].

## Modeling potential giant panda habitat corridors

We used Circuit Model to measure the connectivity among giant panda populations and identify potential habitat corridors [66–69]. We calculated the inverse of the habitat suitability and used it as a resistance surface [67], then we followed the method of McRae et al.(2012) [70]and modeled potential habitat corridors among 33 isolated giant panda populations identified during the 4th National Giant panda Census [49,50,71]. Our hypothesis was that the migration route of giant pandas was a non-directional path, and the populations were concurrently experiencing immigration and emigration [72]. To measure the importance of each corridor in maintaining the overall habitat connectivity pattern, we further used Linkage Mapper to evaluate their centrality following the method developped by Carroll et al.(2012) [73].

## Identifying PHCs for giant pandas

The criteria used to identify PHCs among candidate habitat corridors included biophysical (habitat corridor length and average elevation), biotic (forest coverage and corridor centrality), and anthropogenic (distance to residential areas and nature reserves) metrics [70,74].

We scored each candidate habitat by calculating the degree of dispersion between the actual and the optimal value of its metrics. For example, the known maximum giant panda dispersal distance was 25 km and its seasonal movement distance inside a habitat patch was approximate 2 km [75], based on the above thresholds we scored corridor length for

each candidate habitat corridor with the following rules: 1) any habitat corridor longer than 25 km was removed from the candidate list; 2) candidate habitat corridors shorter than 2 km had the highest score of 1, and habitat corridors with the length of 25 km had the lowest score of 0; and 3) other habitat corridors were scored according to the linear relationship between their lengths and the thresholds on both sides (e.g., 5 km had a score of 0.8, 20 km had a score of 0.2). We admit that this scoring method could be oversimplified at some circumstances, but it provides a generic way and makes ecological sense.

Other metrics and their thresholds include: elevation (the further from the optimal elevational range 1,000 m – 3,000 m, the lower the score, [76]), centrality (the higher the centrality, the higher the score, [73]), forest coverage (the lower the forest coverage, the lower the score, [28]), distance to residential area (the closer the distance to residential area, the lower the score, [77]) and distance to nature reserve (the farther the distance to nature reserve, the lower the score, [78]) (Table 2). Finally, we selected habitat corridors that had the highest overall scores as PHCs (Table 2; [79,80]).

## Results

### Giant panda habitat suitability

The AUC and TSS values of Random Forest (AUC: 0.984 ± 0.003, TSS: 0.906 ± 0.009) and Generalized Boosting Models (AUC: 0.978 ± 0.003, TSS: 0.894 ± 0.008) indicated that the two algorithms had the best discriminative ability (Table S1 in S1 File; Fig 3).

Landscapes with high habitat suitability were in areas that between 1,000–3000 m elevation, had fewer steep slopes (< 20 °) and higher forest coverage (>70%). In addition to environmental constraints, being close to residential areas or roads had a negative association with habitat suitability, and inside or being close to nature reserves had positive association with giant panda habitat suitability (Fig 4).

We identified 55,471 km$^2$ suitable habitat (Fig 4), among which 34,486 km$^2$ (62%) was located inside nature reserves. The area of suitable giant panda habitat in Qinling Mountains was 4,104 km$^2$ (58.7% inside nature reserves), mainly located in the central and western regions inside nature reserves. There was a broad fragmentation pattern for giant panda habitat across six mountain ranges, areas predicted suitable area for giant pandas were divided by roads and other anthropogenic activities. The portion of suitable giant panda habitat outside nature reserves varied among mountain ranges: Qionglai (3,994 km$^2$, 27.2%), Greater Xiangling (906 km$^2$, 69.1%), Lesser Xiangling (937 km$^2$, 54.7%), and Liangshan Mountains (2775 km$^2$, 51.2%) (Fig 4). Landscapes that were potentially suitable for giant pandas but not occupied were mainly located in Shennongjia (1,178 km$^2$) and Bashan Mountains (464 km$^2$).

### Prioritizing PHCs for future habitat corridor establishment

The circuit modeling identified 72 potential giant panda habitat corridors that had a length shorter than 25 km (Fig 5). The potential corridors varied in elevation, length, forest cover, distance to residential areas and nature reserves (Table S2 in S1 File).

Table 2. Scoring standard of priority habitat corridor.

| Limiting factors | Range | Source | Reference |
|---|---|---|---|
| Length (km) | 2–25 | Linkage Mapper | Pan et al. 2014 |
| Elevation (m) | 0–4,122 | ASTER 2009 | Ma et al. 2018 |
| Centrality (Amps) | 1.87–60.83 | Linkage Mapper | Carroll et al. 2012 |
| Forest coverage (%) | 54–100 | Bicheron et al. 2008 | Xu et al. 2006 |
| Distance to residential area (km) | 1–15 | State Forestry Administration 2021 | Shen et al. 2008 |
| Distance to nature reserve (km) | 0–49 | State Forestry Administration 2021 | Vina et al. 2007 |

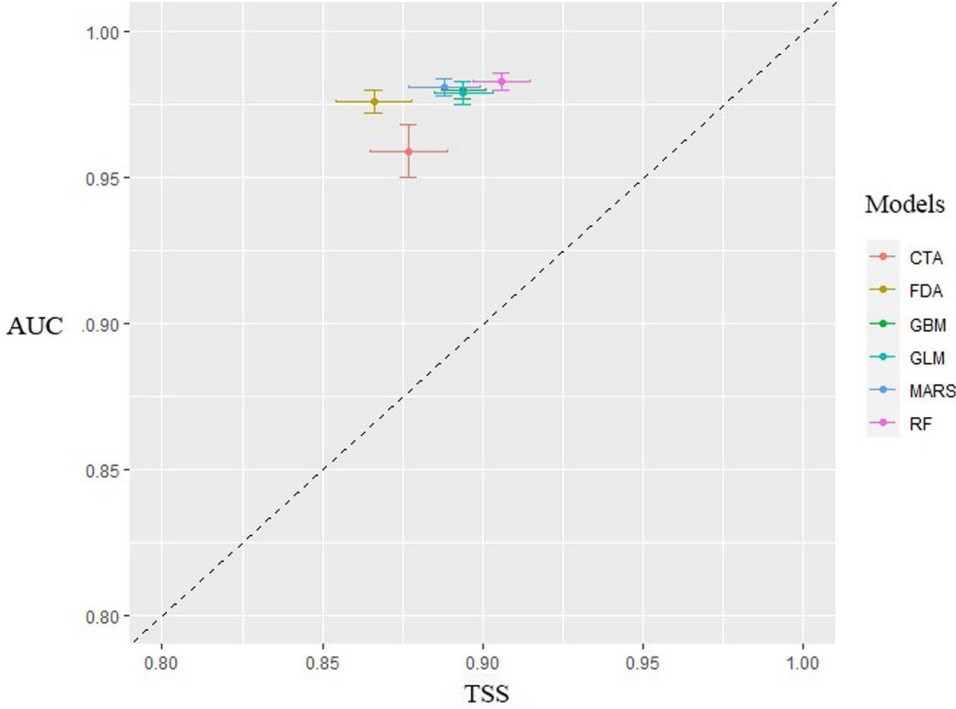

**Fig 3. Cross-validation results using AUC and TSS to compare the performance of different models.** Each crossing symbol indicates the mean and variance of AUC and TSS tests for model replicates. All AUC and TSS result are above dashed line (slope = 1). Abbreviations are as follows. Classification Tree Analysis: CTA, Flexible Discriminant Analysis: FDA, Gradient Boosted Machine Learning: GBM, Generalized Linear Model: GLM, Multivariate Adaptive Regression Splines: MARS, Random Forest Classifier: RF.

We identified six PHCs that merit the highest priority that located within optimal elevation range, had the highest forest cover, linked with multiple populations, and were relatively feasible to manage because they were in or adjacent to current nature reserves and distant to residential areas (Figs 6 and 7).

## Discussion

In this study we proposed a priority framework to select most effective and feasible conservation actions from many candidates based on multiple criteria. We used giant panda habitat corridor establishment as an example to demonstrate how such framework can better support conservation decision-making procedures at regional scale (approximately 27,134 km$^2$) [24,46,48,81].

One important implication of our findings is that, with a few well-designed additions to restore habitat connectivity, the current nature reserves could work as cornerstone of the GPNP. We used biogeographical, environmental, and anthropogenic metrics to select PHCs, and we advocate ecological and socio-economical knowledge are equally important to ensure the long-term effective conservation planning. When nature reserves are established to conserve endangered or vulnerable species, there used to be an implicit understanding that the conservation effort will inevitably conflicts with economic development [82]. As a result, many conservation efforts were targeted at eliminating human footprint which resulted in major resettlement and severe conflict between local communities and nature reserves [7,83,84]. In this case, all PHCs were adjacent to nature reserves but relatively far from residential areas, making it easier to allocate existing conservation resources and avoid much conflict [85]. The rest habitat corridors, especially the ones that met more than one criteria with high scores, can further help species' long-term survival in the next stage [86]. In addition to giant panda

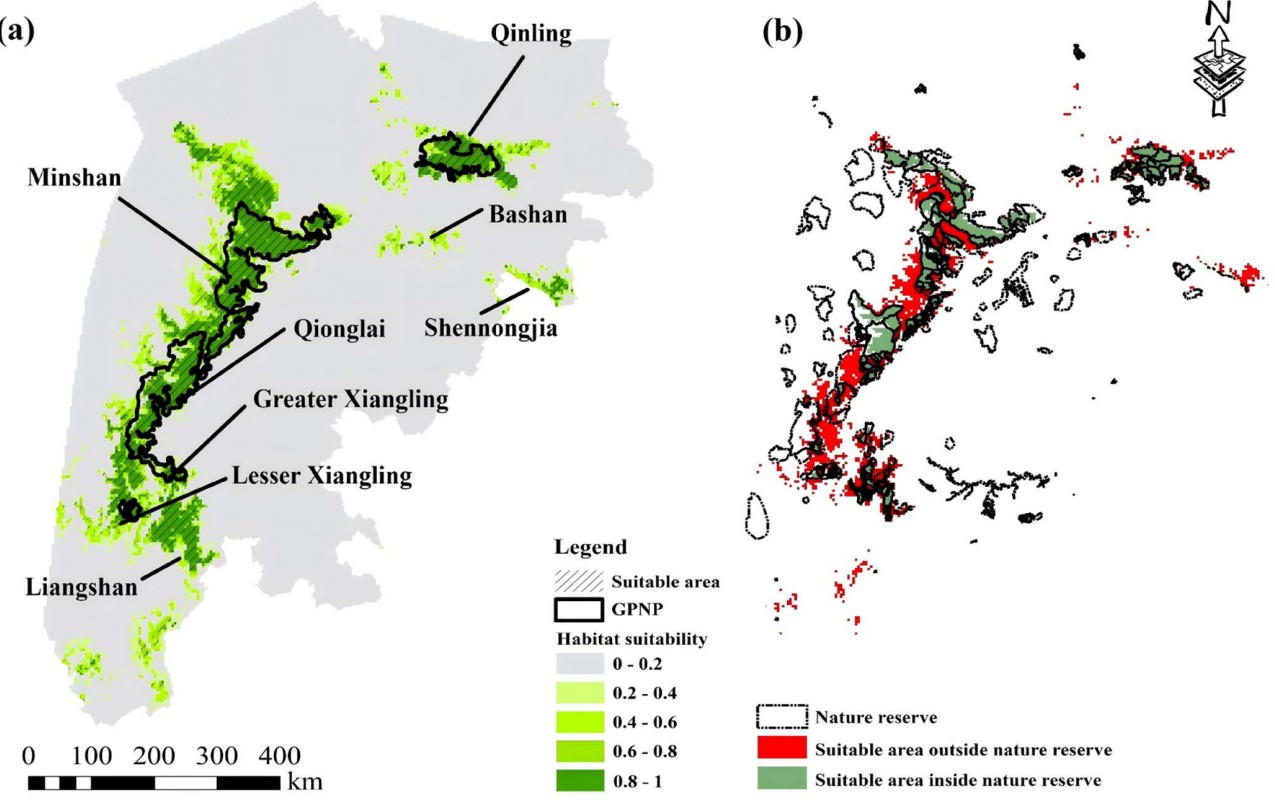

**Fig 4. Giant panda habitat suitability around GPNP (a) and giant panda habitat in (green) and outside (red) nature reserves (b).** There was a broad fragmentation pattern for giant panda habitat across six mountain ranges, areas predicted suitable area for giant pandas were in patches with higher forest coverage and less intensive anthropogenic activities. The area of suitable area for giant pandas within nature reserves is 34,486 km², while the suitable area outside nature reserves is 20,984 km².

conservation, we believe the framework we proposed is relatively simple, the analytical tools and interdisciplinary criterions are generally suited to more tractable systems.

### Reintroduction and expansion

We believe our prioritizing framework also provides a tool to support other conservation initiatives, one example is to identify future reintroduction sites. We advocate that the PHCs we identified in this study should have the priority in future giant panda reintroduction, to promoting giant panda movement among isolated population. Interesting to us, the model predictions identified large areas of potential suitable habitats for giant pandas in the Bashan Mountains, Shennongjia, and the western Qinling Mountains, which were once inhabited by giant pandas in the 19th century, and have become future habitat for existing pandas as a result of forest restoration. Historically, giant panda population in these regions were extirpated as late as late 19th century [75,87]. The reasons for the recent extirpation were habitat degradation caused by human expansion and the massive bamboo flowering afterwards (*Fargesia spathacea, F. murielae, and Yushania confusa*) [88]. Recent studies indicated that the National Forest Protection Project had successfully restored the forest and bamboo understory in these regions, the conversion of grassland and cropland to forestland significantly increased the connectivity of giant panda habitats, and strict logging and poaching control were well enforced [50,88,89]. Besides PHCs, regions such as Bashan and western Qinling Mountains could be used as candidate sites to expand species distribution to its historical range.

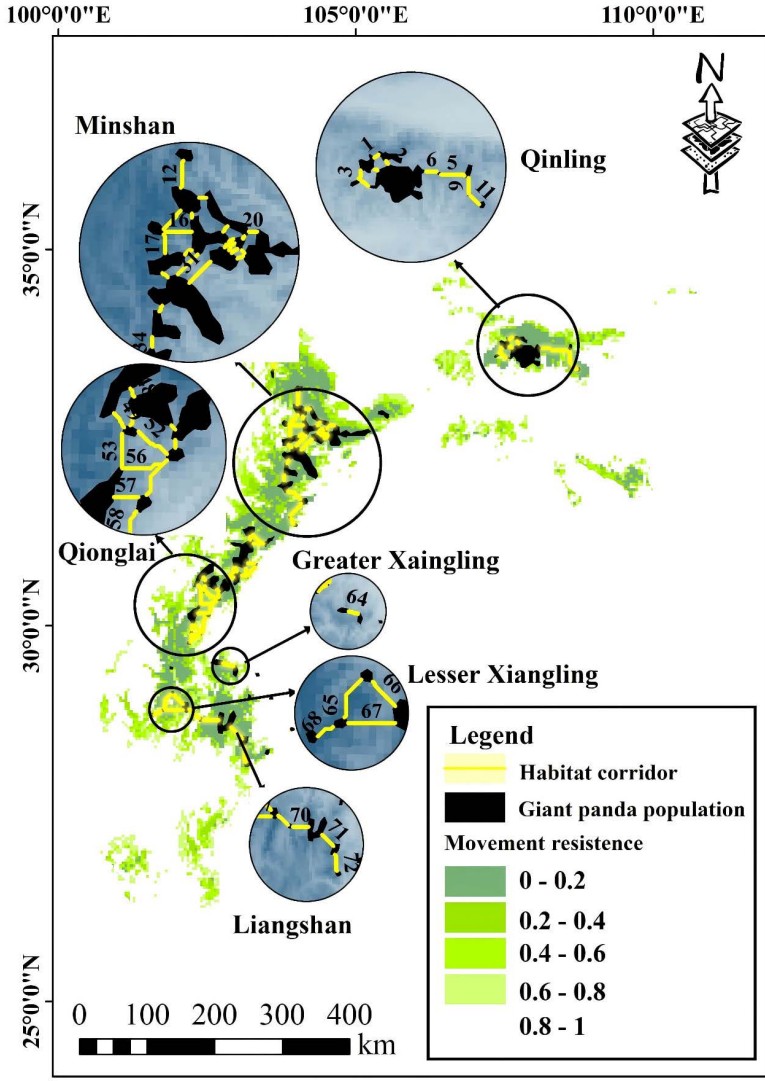

**Fig 5. The circuitscape model identified 72 potential habitat corridors among giant panda populations.** The potential habitat corridors connecting different populations of giant pandas, varied in elevation, length, forest cover, distance to residential areas and nature reserves.

## Habitat corridors selection

While the number of nature reserves in China has reached up to 2,538 with a total area of 1.43 million km², such large geographic coverage does not necessarily imply adequate protection, especially when major residential areas and roads are acting as barriers when restoring habitat connectivity beyond nature reserve boundaries [90–92]. In this study, we identified six highest priority corridors centered on Minshan mountain, where had corresponding highest relative abundance metapopulation and supported as a gene pool needed to communicate with other small populations [93]. These conservation efforts were proved to be beneficial in improving the connectivity of giant panda populations, enhancing gene flow, and reducing inbreeding [94]. Previous studies on giant panda and other large mammals have described a broad spectrum of negative effects of farming and grazing even inside nature reserves [48,95,96], and we further revealed the lack of large-scale conservation planning outside nature reserves had restricted the effectiveness of the conservation network.

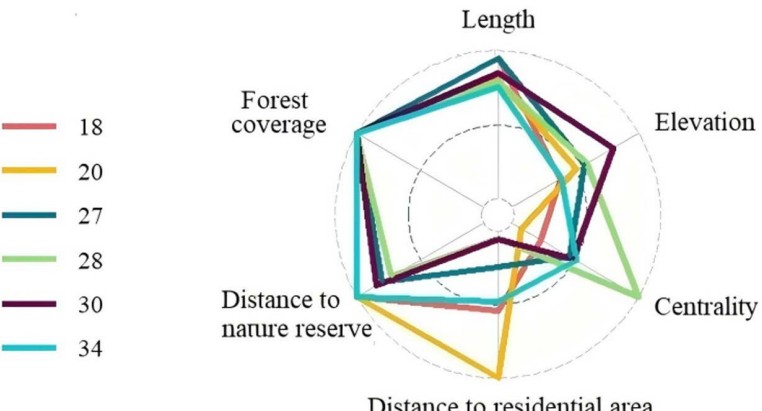

**Fig 6. Metrics score of the six PHCs.** PHCs were located within optimal elevation range, had the highest forest cover, linked with multiple populations, and were relatively feasible to manage because they were in or adjacent to current nature reserves and distant to residential areas.

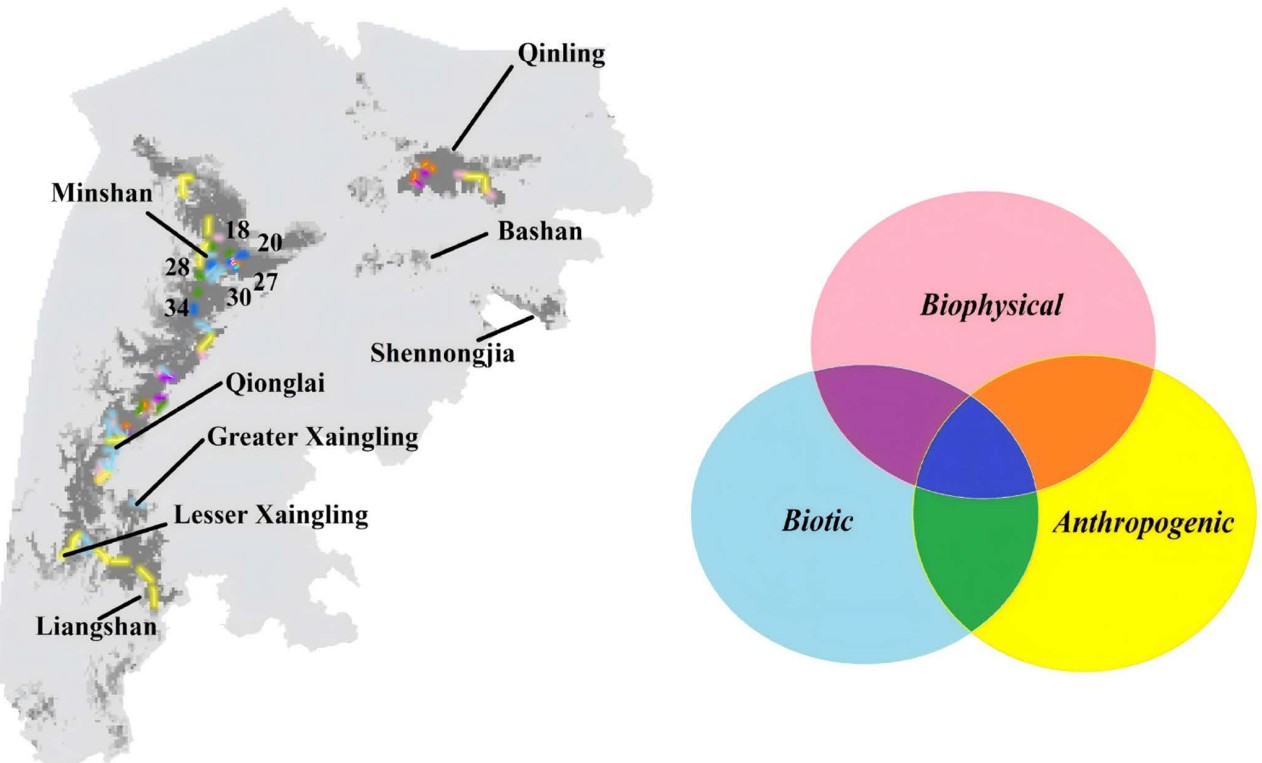

**Fig 7. Location of Priority Habitat Corridors (PHC).** Besides PHCs (darker blue), there were candidate habitat corridors that met two of the three criteria (pink, green, and orange) that can be considered moderate important in future conservation planning. Single colors indicate locations with potential habitat corridors score high in one criterion, overlapping colors indicate habitat corridors according to meet more than one criterion, as colors as shown in the Venn diagrams.

Although the importance of habitat corridors has been consistently emphasized [97,98], large scale corridor planning had been few successful cases for many species include giant pandas [20]. Chinese government had established giant panda habitat corridors including Huangtuliang corridors (located between Wanglang and Baishuijiang nature reserves), Tudiling corridors (located in Jiudingshan nature reserve), Niba Mountain corridors (connected Daxiangling and Qionglai Mountains populations, where separated by roads and rivers, as well as by human settlements and cropland areas, into four main habitat blocks), Tuowu Mountain corridors (joined two giant panda populations in Xiaoxiangling Mountains) and 108 National Highway of Qinling Tunnel (linked Zhouzhi and Guanyinshan nature reserve) giant panda habitat corridors [28]. It is worth noting that PHCs (Huangtuliang, Tudiling and 108 National Highway of Qinling Tunnel) identified by our framework had been reported the migration of giant pandas [99,100]. It is noteworthy we removed any potential habitat corridors that were longer than 25 km, because overlong habitat corridor establishment could not promote species migration but may have opposite effects.

## Conservation recommendations

We first had a result like previous studies that there were 72 habitat corridors across three provinces in six mountain ranges, but such task seems an impossible mission for any conservation agenda [101] . By moving a step forward to translate the habitat corridor metrics into explicit objectives, our priority framework reduced the number of candidate locations by 82% and suggested specific actions for each PHCs to mitigate the remaining negative impact of limiting factors. We believe the framework has general value in giant panda and other conservation decision-making procedures, where there limited resources need to be allocated to the most critical areas.

Based on our results, we suggest three actions in addition to giant panda habitat corridor establishment, including: (1) establish new nature reserves in conservation GAPs to cover highly suitable habitat outside nature reserves, especially in Qionglai and Liangshan Mountains where there were inadequate nature reserve coverage; (2) expand current nature reserves to cover low-elevation area, especially in Liangshan and Greater Xiangling Mountains where giant panda are distributed in lower elevation; and (3) investigate habitat needs for other species and expand the conservation effort to a broad spectrum of species [5,26,81]. We advocate using the establishment of the new GPNP as a chance to better allocate limited resources on urgent conservation needs [10,11].

## Supporting information

**S1 File.  S1 Table**. TSS and AUC in each model in the ensemble model ± standard error. **S2 Table**. The length, elevation and other characteristics of potential giant panda habitat corridors.
(DOCX)

## Acknowledgments

We thank the staff of the Shaanxi Forestry Department who helped us in collecting environment information. We are very grateful to Dr. McShea for his advice in conducting this research. The funders had no role in study design, data collection and analysis, decision to publish, or preparation of the manuscript.

## Author contributions

**Conceptualization:** Fang Wang.

**Formal analysis:** Yue Weng, Qianqian Zhao, Xiangxue Hu, Yihan Wang, Zhuojin Zhang, Fang Wang.

**Investigation:** Yixin Diao, Yue Weng, Qianqian Zhao, Xiaofeng Zhang, Bojian Gu.

**Methodology:** Yixin Diao, Fang Wang.

**Visualization:** Yixin Diao.

**Writing – original draft:** Yixin Diao, Fang Wang.

**Writing – review & editing:** Yixin Diao, Xiangxue Hu, Fang Wang.

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
