## [Decision Letter · Decision Letter 0]

Dear Dr. Diao,

Thank you for submitting your manuscript to PLOS ONE. After careful consideration, we feel that it has merit but does not fully meet PLOS ONE’s publication criteria as it currently stands. Therefore, we invite you to submit a revised version of the manuscript that addresses the points raised during the review process.

This manuscript proposes an interesting framework to identify Priority Habitat Corridors for giant pandas in the newly established Giant Panda National Park, using integrated modeling with biophysical, biotic, and anthropogenic criteria. By providing a standardized methodology to prioritize conservation actions, this work offers some valuable support for future management of giant pandas and other wildlife species, helping improve the allocation of limited resources to the most critical areas for maximum conservation impact. While this study will be of interest to bear ecologists and panda conservation managers, there are some issues with this manuscript that preclude its publication in its current state.

Consequently, major changes are required before the manuscript can be further considered. Both reviewers provide insightful comments and suggestions that will help guide the revision process to improve this manuscript. The modeling framework is not particularly innovative and needs further detail and justification. There needs to be stronger integration of results with known genetic distribution of giant pandas and more discussion on landscape changes since earlier corridor predictions. Both reviewers also provide several additional references for the authors to consider incorporating into their manuscript, especially with regard to recent developments in corridor modeling.

We look forward to receiving your revised manuscript.

Kind regards,

James K. Sheppard

Academic Editor

PLOS ONE

“We thank the staff of the Shaanxi Forestry Department who helped us in collecting environment information. We are very grateful to Dr. McShea for his advice in conducting this research. The project was supported by the National Natural Science Foundation of China (31971537) and Qiankehe [ZK] (092).”

“We thank the staff of the Shaanxi Forestry Department who helped us in collecting environment information. We are very grateful to Dr. McShea for his advice in conducting this research. The project was supported by the National Natural Science Foundation of China (31971537) and Qiankehe [ZK] (092).”

“We thank the staff of the Shaanxi Forestry Department who helped us in collecting environment information. We are very grateful to Dr. McShea for his advice in conducting this research. The project was supported by the National Natural Science Foundation of China (31971537) and Qiankehe [ZK] (092).”

4. We note that Figures 2,4,5 and 7 in your submission contain [map/satellite] images which may be copyrighted. All PLOS content is published under the Creative Commons Attribution License (CC BY 4.0), which means that the manuscript, images, and Supporting Information files will be freely available online, and any third party is permitted to access, download, copy, distribute, and use these materials in any way, even commercially, with proper attribution. For these reasons, we cannot publish previously copyrighted maps or satellite images created using proprietary data, such as Google software (Google Maps, Street View, and Earth). For more information, see our copyright guidelines: http://journals.plos.org/plosone/s/licenses-and-copyright.

a. You may seek permission from the original copyright holder of Figures 2,4,5 and 7 to publish the content specifically under the CC BY 4.0 license. 

5. We notice that your supplementary tables are included in the manuscript file. Please remove them and upload them with the file type 'Supporting Information'. Please ensure that each Supporting Information file has a legend listed in the manuscript after the references list.

Additional Editor Comments:

This manuscript proposes an interesting framework to identify Priority Habitat Corridors for giant pandas in the newly established Giant Panda National Park, using integrated modeling with biophysical, biotic, and anthropogenic criteria. By providing a standardized methodology to prioritize conservation actions, this work offers some valuable support for future management of giant pandas and other wildlife species, helping to allocate limited resources to the most critical areas for maximum conservation impact. While this study will be of interest to bear ecologists and panda conservation managers, there are some issues with this manuscript that preclude its publication in its current state.

Consequently, major changes are required before the manuscript can be further considered. Both Reviewers provide insightful comments and suggestions that will help guide the revision process to improve this manuscript. The modeling framework is not particularly innovative and needs further detail and justification. There needs to be stronger integration of results with known genetic distribution of giant pandas and more discussion on landscape changes since earlier corridor predictions. Both reviewers also provide several additional references for the authors to consider incorporating into their manuscript, especially with regard to recent developments in corridor modeling.

Reviewers' comments:

Reviewer's Responses to Questions

**Comments to the Author**

1. Is the manuscript technically sound, and do the data support the conclusions?

Reviewer #1: Yes

Reviewer #2: No

2. Has the statistical analysis been performed appropriately and rigorously?

Reviewer #1: Yes

Reviewer #2: I Don't Know

3. Have the authors made all data underlying the findings in their manuscript fully available?

Reviewer #1: Yes

Reviewer #2: No

4. Is the manuscript presented in an intelligible fashion and written in standard English?

Reviewer #1: Yes

Reviewer #2: Yes

Reviewer #1: I thought this was an interesting paper and it was well written. I think it is a pertinent topic. I have no major concerns. I have some minor suggestions for improvement below.

In general I think there could be a stronger integration of Results/Implications of this study with what we know about genetic distribution of the species (in the Discussion section specifically). The citation below would be great to explore in detail, and I think it appears in your Reference list but is not cited in the paper (Unless I missed it?)

Wang, Meng, Guiming Wang, Guangping Huang, Andy Kouba, Ronald R. Swaisgood, Wenliang Zhou, Yibo Hu, Yonggang Nie, and Fuwen Wei. "Habitat connectivity drives panda recovery." Current Biology 34, no. 17 (2024): 3894-3904.

I think it might be interesting to touch on more about how things have changed in the landscape since earlier corridor predictions. I wonder about comparing to this paper below from around 2 decades ago (although this is Qionglai only)

Xu, Weihua, Zhiyun Ouyang, Andrés Viña, Hua Zheng, Jianguo Liu, and Yi Xiao. "Designing a conservation plan for protecting the habitat for giant pandas in the Qionglai mountain range, China." Diversity and Distributions 12, no. 5 (2006): 610-619.

I also think it might be interesting to talk more about what your recommendations mean in light of what has actually been done (or more importantly not done) with respect to actual corridor construction or restoration to date. Maybe this paper can be a good citation to consider.

Kang, Dongwei. "A review of the habitat restoration of giant pandas from 2012 to 2021: Research topics and advances." Science of the Total Environment 852 (2022): 158207.

Another reference to consider if helpful for adding context to the Discussion:

Hu, Lu, Bin Feng, Jindong Zhang, Xin Dong, Junfeng Tang, Caiquan Zhou, Dunwu Qi, and Wenke Bai. "Impacts of land-use change on the habitat suitability and connectivity of giant panda." Global Ecology and Conservation (2024): e03019.

Other

Line 368- capitalize China

Line 375-376- This sentence was hard to understand and should be reworded or fleshed out in more detail

Line 399- "giant pandas are distributed in lower elevations"

Reviewer #2: Thank you for the opportunity to review this work. I appreciate its motivation, and there seem to be some intriguing potential findings.

While I think there is a worthy publication in this work, I do not believe this manuscript is yet where it needs to be - to be published. I have provided some major and minor comments that I hope you can take and help transition this into a more complete and finalized product.

Major:

There is no new framework provided here. Much fundamental corridor literature has not been cited, including multiple decision science-based corridor frameworks. This includes from the 1980s with ordinal rankings to the following:

Bond ML, Bradley CM, Kiffner C, Morrison TA, Lee DE. A multi-method approach to delineate and validate migratory corridors. Landscape Ecol. 2017;32:1705–21.

Osipova L, Okello MM, Njumbi SJ, Ngene S, Western D, Hayward MW, et al. Validating movement corridors for African elephants predicted from resistance-based landscape connectivity models. Landscape Ecol. 2019;34:865–78.

Barnett K, Belote RT. Modeling an aspirational connected network of protected areas across North America. Ecol Appl. 2021;31:e02387.

Dinerstein E, Joshi AR, Vynne C, Lee ATL, Pharand-Deschênes F, França M, et al. A “Global Safety Net” to reverse biodiversity loss and stabilize Earth’s climate. Sci Adv. 2020;6:eabb2824.

Cameron DR, Schloss CA, Theobald DM, Morrison SA. A framework to select strategies for conserving and restoring habitat connectivity in complex landscapes. Conserv Sci Pract. 2022;4:e12698.

Brennan A, Naidoo R, Greenstreet L, Mehrabi Z, Ramankutty N, Kremen C. Functional connectivity of the world’s protected areas. Science. 2022;376:1101–4.

aruemon Tantipisanuh, Somporn Phakpian, Pornpimon Tangtorwongsakul, Supagit Vinitpornsawan, Dusit Ngoprasert, Identifying wildlife corridors to restore population connectivity: An integration approach involving multiple data sources, Global Ecology and Conservation, Volume 53, 2024,

https://doi.org/10.1016/j.gecco.2024.e03015.

Generally, this manuscript lacks methodological detail and justification for the chosen methods.

Many models were considered for the final ensemble without any justification… such as diversity or representation of modeling methods. There is a circular nature to using covariates in habitat suitability, which are then incorporated to some extent into the elimination process. The modeling processes were not described even in the supplement.

Minor:

There were many minor errors in writing and grammar, mostly with tense, article placement, and singular/plural.

**Do you want your identity to be public for this peer review?** For information about this choice, including consent withdrawal, please see our Privacy Policy

Reviewer #1: No

Reviewer #2: No

---

## [Author Response · Author response to Decision Letter 1]

20 Apr 2025

Responses to reviewers’ comments

Dear Editors and Reviewers,

We would like to thank you cordially for your thorough and constructive comments concerning our manuscript entitled " Using multiple criteria for redesigning habitat corridor plans for giant pandas” (PONE-D-24-51006). We do feel that those comments are all valuable and helpful for us to improve this manuscript. With the guidance of the comments, we made a quite extensive revision with details as following:

Responses to the comments from Editor (Relevant text changes made in yellow in the revised manuscript)

1. When submitting your revision, we need you to address these additional requirements. Please ensure that your manuscript meets PLOS ONE's style requirements, including those for file naming.

Response: Thank the editor for this comment. We have revised the manuscript using the standard format suggested by the editor.

Response: Thanks for your valuable suggestion. We added statement in the acknowledgements: "The funders had no role in study design, data collection and analysis, decision to publish, or preparation of the manuscript."

Response: We appreciate the editor’s comments. We deleted funding information in the acknowledgements.

4. You may seek permission from the original copyright holder of Figures 2,4,5 and 7 to publish the content specifically under the CC BY 4.0 license.

Response: Thank the editor for this comment. Figure 2 was generated based on the location descriptions of 5 national parks published by https://www.gov.cn/, using the public database OpenStreetMap (abbreviated OSM) that allows CC BY 4.0. Figure 4 and 5 were drawn in R studio packages exclusive of copyrighted maps or satellite images created using proprietary data. We confirmed that all images were self-drawn and correspond with CC BY 4.0.

5. We notice that your supplementary tables are included in the manuscript file. Please remove them and upload them with the file type 'Supporting Information'. Please ensure that each Supporting Information file has a legend listed in the manuscript after the references list.

Response: Thanks for your suggestion. We have removed supplementary tables from the manuscript and make sure that each Supporting Information file has a legend listed in the manuscript after the references list.

Responses to the comments from Reviewer #1 (Relevant text changes made in green in the revised manuscript)

Main comments: I thought this was an interesting paper and it was well written. I think it is a pertinent topic. I have no major concerns. I have some minor suggestions for improvement below.

Response: We really appreciate your valuable comments and great efforts on guiding our study. We have tried to address the comments as following.

Comment 1: In general I think there could be a stronger integration of Results/Implications of this study with what we know about genetic distribution of the species (in the Discussion section specifically). The citation below would be great to explore in detail, and I think it appears in your Reference list but is not cited in the paper (Unless I missed it?)

Response: Thank the reviewer for this comment. We revised more genetic distribution and population information in Discussion section (L 328-330).

Comment 2: I think it might be interesting to touch on more about how things have changed in the landscape since earlier corridor predictions. I wonder about comparing to this paper below from around 2 decades ago (although this is Qionglai only)

Response: Thanks for your valuable suggestion. We cited Xu et al. (2006) in our Method section (L 205), we further talked about the benefits of existing corridors for different populations, as well as for populations isolated by roads in the Qionglai Mountains (L 337-346).

Comment 3: I also think it might be interesting to talk more about what your recommendations mean in light of what has actually been done (or more importantly not done) with respect to actual corridor construction or restoration to date. Maybe this paper can be a good citation to consider.

Response: We appreciate the reviewer’s comments. As far as we know, Chinese government had established many giant panda habitat corridors to link geographically isolated populations. It is worth noting that PHCs (Huangtuliang, Tudiling and 108 National Highway of Qinling Tunnel) identified by our framework had been reported the migration of giant pandas (Xiao et al., 2021; Kang, 2022). The remaining habitat priority corridors should also be linked as soon as possible (L 335-351).

Comment 4: Another reference to consider if helpful for adding context to the Discussion:

Hu, Lu, Bin Feng, Jindong Zhang, Xin Dong, Junfeng Tang, Caiquan Zhou, Dunwu Qi, and Wenke Bai. "Impacts of land-use change on the habitat suitability and connectivity of giant panda." Global Ecology and Conservation (2024): e03019.

Response: Thank the reviewer for this comment. We added impacts of land-use change on the habitat suitability and connectivity of giant panda to the discussion section.

Comment 5:

Line 368- capitalize China

Line 375-376- This sentence was hard to understand and should be reworded or fleshed out in more detail

Line 399- "giant pandas are distributed in lower elevations

Response: Thank the reviewer for this comment. We followed the recommendations and revised the manuscript.

The following citations were added to the revised manuscript:

Wang M, Wang G, Huang G, Kouba A, Swaisgood RR, Zhou W, Hu Y, Nie Y, Wei F. Habitat connectivity drives panda recovery. Curr Biol. 2024 Sep 9;34(17):3894-3904.e3.

Xu, W., Ouyang, Z., Viña, A., Zheng, H., Liu, J., Xiao, Y., 2006. Designing a conservation plan for protecting the habitat for giant pandas in the Qionglai mountain range, China. Divers. Distrib. 12, 610–619.

Hu, L., Feng, B., Zhang, J., Dong, X., Tang, J., Zhou, C., Qi, D., Bai, W., 2024. Impacts of land-use change on the habitat suitability and connectivity of giant panda. Global Ecology and Conservation, e03019.

Fontoura, L., D’agata, S., Gamoyo, M., Barneche, D.R., Luiz, O.J., Madin, E.M.P., Eggertsen, L., Maina, J.M., 2022. Protecting connectivity promotes successful biodiversity and fisheries conservation. Science 375, 336–340.

Fricke, E.C., Ordonez, A., Rogers, H.S., Svenning, J.-C., 2022. The effects of defaunation on plants’ capacity to track climate change. Science 375, 210–214.

Xiao, L., Xie, J., Zhang, C., 2021. Construction of giant panda exchange habitat corridors in Giant Panda National Park. New China News Agency. http://www.news.cn/politics/2021-12/02/c_1128125251.html

Kang, D, 2022. A review of the habitat restoration of giant pandas from 2012 to 2021: Research topics and advances. Sci. Total Environ. 852, 158207.

Responses to the comments from Reviewer #2 (Relevant text changes made in blue in the revised manuscript)

Reviewer #2:

Main comments: Thank you for the opportunity to review this work. I appreciate its motivation, and there seem to be some intriguing potential findings.

While I think there is a worthy publication in this work, I do not believe this manuscript is yet where it needs to be - to be published. I have provided some major and minor comments that I hope you can take and help transition this into a more complete and finalized product.

Response: We are very grateful for your valuable comments and the great effort you put into guiding our research. We have attempted to respond to these comments as follows.

Comment 1: There is no new framework provided here. Much fundamental corridor literature has not been cited, including multiple decision science-based corridor frameworks.

Response: We appreciate reviewer’s insight, and have revised our manuscript to better explain our framework. We read the relevant literature and found that they mainly focus on species and the environment data with less consideration of socio-economic topics. We have added references above in the section of biophysical and biotic, anthropogenic criteria as filters to select PHCs (L 87-95).

Comment 2: Generally, this manuscript lacks methodological detail and justification for the chosen methods.

Many models were considered for the final ensemble without any justification… such as diversity or representation of modeling methods. There is a circular nature to using covariates in habitat suitability, which are then incorporated to some extent into the elimination process. The modeling processes were not described even in the supplement.

Response: Thank the reviewer for this comment. For the issue of methodological detail and justification for the chosen methods, we chose 9 environment variables based on VIF when modelling habitat suitability (L 146-150; table 1). We randomly selected 70% giant panda locations for model training and used the remaining 30% to assess the discriminative ability of different model algorithms followed Hunter et al. (2010). Furthermore, we have identified key ecological factors and constructed least-cost and circuit models to identify potential movement pathways affecting giant pandas in the current researches and (Bu et al., 2021;Wang al., 2014; Wang et al., 2015; Wang et al., 2018).

Comment3: There were many minor errors in writing and grammar, mostly with tense, article placement, and singular/plural.

Response: Thank the reviewer for this comment. We checked and revised the manuscript

The following citations were added to the revised manuscript:

Bond, M. L., Bradley, C. M., Kiffner, C., Morrison, T. A., Lee, D. E., 2017. A multi-method approach to delineate and validate migratory corridors. Landscape Ecol. 32, 1705-1721.

Bu, H., McShea, W. J., Wang, D., Wang, F., Chen, Y., Gu, X., Li, S., 2021. Not all forests are alike: The role of commercial forest in the conservation of landscape connectivity for the giant panda. Landscape Ecol. 36(9), 2549-2564.

Osipova, L., Okello, M. M., Njumbi, S. J., Ngene, S., Western, D., Hayward, M. W., Balkenhol, N., 2019. Validating movement corridors for African elephants predicted from resistance-based landscape connectivity models. Landscape Ecol. 34, 865-878.

Barnett, K., Belote, R. T., 2021. Modeling an aspirational connected network of protected areas across North America. Ecol. Appl. 31, e02387.

Dinerstein, E., Joshi, A. R., Vynne, C., Lee, A. T., Pharand-Deschênes, F., França, M., Olson, D., 2020. A “Global Safety Net” to reverse biodiversity loss and stabilize Earth’s climate. Sci Adv. 6, eabb2824.

Cameron, D. R., Schloss, C. A., Theobald, D. M., Morrison, S. A., 2022. A framework to select strategies for conserving and restoring habitat connectivity in complex landscapes. Conserv. Sci. Pract. 4, e12698.

Barnett, K., Belote, R. T., 2021. Modeling an aspirational connected network of protected areas across North America. Ecol. Appl. 31, e02387.

Tantipisanuh, N., Phakpian, S., Tangtorwongsakul, P., Vinitpornsawan, S., Ngoprasert, D., 2024. Identifying wildlife corridors to restore population connectivity: An integration approach involving multiple data sources. Glob. Ecol. Conserv. e03015.

Wang, F., Zhao, Q., McShea, W.J., Songer, M., Huang, Q., Zhang, X., Zhou, L., 2018. Incorporating biotic interactions reveals potential climate tolerance of giant pandas. Conserv. Lett. 11, e12592.

Wang, F., McShea, W.J., Wang, D., Li, S., Zhao, Q., Wang, H., Lu, Z., 2014. Evaluating landscape options for corridor restoration between giant panda reserves. PLoS One 9, 1–9.

---

## [Editor Report · Decision Letter 1]

Using multiple criteria for redesigning habitat corridor plans for giant pandas

PONE-D-24-51006R1

Dear Dr. Wang,

We’re pleased to inform you that your manuscript has been judged scientifically suitable for publication and will be formally accepted for publication once it meets all outstanding technical requirements.

Kind regards,

James K. Sheppard

Academic Editor

PLOS ONE
---

## [Editor Report · Acceptance letter]

PONE-D-24-51006R1

PLOS ONE

Dear Dr. Wang,

I'm pleased to inform you that your manuscript has been deemed suitable for publication in PLOS ONE. Congratulations! Your manuscript is now being handed over to our production team.

Kind regards,

on behalf of

Dr. James K. Sheppard

Academic Editor

PLOS ONE